# Masculinization of Adult *Gambusia holbrooki*: A Case of Recapitulation of Protogyny in a Gonochorist?

**DOI:** 10.3390/biology11050694

**Published:** 2022-04-30

**Authors:** Ngoc Kim Tran, Tzu Nin Kwan, John Purser, Jawahar G. Patil

**Affiliations:** 1Fisheries and Aquaculture Centre, Institute for Marine and Antarctic Studies, University of Tasmania, Launceston, TAS 7250, Australia; trankn@utas.edu.au (N.K.T.); john.purser@utas.edu.au (J.P.); 2Department of Aquaculture, Faculty of Agriculture and Natural Resources, An Giang University, 18 Ung Van Khiem Street, Long Xuyen City 880000, Vietnam; 3Fisheries and Aquaculture Centre, Institute for Marine and Antarctic Studies, University of Tasmania, Taroona, TAS 7053, Australia; tzu.kwan@utas.edu.au; 4Inland Fisheries Services, New Norfolk, TAS 7140, Australia

**Keywords:** eastern mosquitofish, 17α-Methyltestosterone, Anti-Müllerian Hormone (*amh*) gene, gonopodial development, testicular tissue

## Abstract

**Simple Summary:**

Although gonochoristic fish express one sex or the other in adulthood, some adults display traits similar to those of their opposite sex, suggesting plasticity similar to hermaphrodites. To fully test this potential in the eastern mosquitofish, a gonochorist, two adult stages of females were fed 17α-Methyltestosterone incorporated feed (0–200 mg/kg diet) for 50 days. The hormone (particularly at 50 mg/kg diet) stimulated the formation of complex male copulatory structures and testicular tissue as well as upregulated expression of Anti-Müllerian Hormone gene and altered the behavior from females to males. Collectively, we infer that *Gambusia holbrooki* retains potential for sex reversal at adulthood, similar to what occurs naturally in hermaphroditic fish and can provide an ideal system to investigate these mechanisms in a regulated fashion.

**Abstract:**

17α-Methyltestosterone (MT) is a synthetic steroid that has been widely used to masculinize many fish species when administered early during larval development, however, reports on its efficacy on adults is limited. To this end, this study investigated the efficacy of MT in the masculinization of the eastern mosquitofish (*G. holbrooki*) at two adult stages (maiden and repeat gravid females). The treated females were fed control or respective MT incorporated feed (0–200 mg/kg diet) for 50 days. Effects of the hormone on secondary sexual characteristics, internal gonad morphology, expression of the Anti-Müllerian Hormone (*amh*) gene and sexual behavior of the treated females were investigated. The results showed that MT at the dose of 50 mg/kg feed stimulated secondary sexual character development, upregulated expression of *amh*, formation of testicular tissue and a shift in the behavior similar to those of normal males, prominently so in treated maiden gravid females. Post-treatment, long-term observations indicated that only two masculinized females reverted back to being females and gave birth to young. Induction of masculinizing effects in most individuals suggests that the sexual phenotype of this species appears to be highly plastic with potential to sex reverse at adulthood. This in combination with its small size and short reproductive cycle could provide an ideal system to explore the mechanisms of sequential hermaphroditism in fish and contribute to genetic control of this pest fish.

## 1. Introduction

Teleosts exhibit a wide range of sexual patterns, namely gonochorism, unisexualism and hermaphroditism [1]. In gonochoristic fish, there are two major pathways—primary and secondary gonochores [2]. In the former, the sex differentiation process follows classical pathways of genetic sex determination [3] whereas the latter exhibit a non-functional intersexual phase leading to delayed differentiation [2]. In contrast, male and female reproductive functions in hermaphrodites are expressed within a single individual, occurring in 1% of vertebrate species of which nearly all are fish [4]. Functional hermaphroditism, either simultaneous, sequential or serial [3] has been confirmed in more than 450 species in 41 families of 17 teleost orders [4]. Within the live-bearing family Poeciliidae, which consists of 293 species [5], only *Xiphophorus* (swordtails) has been proposed to exhibit hermaphroditism (reviewed in [6]). However, the reasons and mechanisms that confer such sexual plasticity in adults are relatively less known.

Interestingly, masculinization (i.e., conversion of genetic females to functional males) in gonochoristic fish can be achieved by exposure to exogenous androgens or by blocking aromatase activity during early larval development [7,8]. The synthetic androgen, 17α-Methyltestosterone (MT), is the most widely used hormone for masculinization of fish for both aquaculture and ornamental fish applications [9,10]. In poeciliids, MT has been used to masculinize newborn guppy [11,12,13], swordtails [14] and black molly [15]. In contrast, no studies have demonstrated MT treatment to embryonic stages via gravid females except for the German report of Dzwillo (1966, as cited in [11]). Although, MT dietary administration to adult females stimulates the development of male secondary sexual characteristics, e.g., the growth of sword extension [16,17]; the development of male secondary sexual coloration [18], none have resulted in functional sex reversal. Similarly, eco-toxicological reports on the effects of endocrine disrupting chemicals (EDCs) on adults are also limited to masculinization of secondary sexual characters, e.g., [19].

In poeciliid fish, males and females are easily distinguishable by secondary sexual organs. Males possess a gonopodium, a differentiated anal fin formed by the elongation and modification of anal fin rays 3, 4 and 5 [20], which are driven by androgenic hormones produced by the testis [21,22]. In contrast, mature females have an undifferentiated anal fin and possess a gravid spot which is an excellent marker to identify mature and brooding females [23], thus making them excellent systems to study sexual perturbations. For example, exposure to androgenic compounds in *G. affinis*, a sister species of *G. holbrooki*, induces the development of gonopodium-like structures in treated females mimicking EDC effects in the wild [20,24].

Most gonochoristic teleosts develop either testes or ovaries, with some species displaying intersex characteristics, a signature effect of exposure to EDCs [25]. Occurrence of intersex fish has been reported in three species of euryhaline mullet *Chelon labrosus*, *Liza aurata* and *Mugil cephalus* [26], and in freshwater catfish *Steindachneridition parahybae* [27] living in polluted waters. Interestingly, a common condition found in intersex fish is the presence of oocytes in testes, referred to as ‘testicular oocytes’ (reviewed in [28]). In contrast, the intersex condition where male tissue (e.g., spermatocytes) is present in mature ovaries, i.e., ovotestis is less commonly reported and seems to be caused by androgenic or anti-estrogenic exposure [28]. This condition has been reported in female Japanese medaka *Oryzias latipes* [29], guppies, *Poecilia reticulata* [11], gynogenetic brook trout *Salvelinus fontinalis* [30] and rainbow trout *Oncorhynchus mykiss* [31] when treated or fed with androgenic compounds during early larval development. However, no studies have reported this condition in adult fish treated with androgens.

Increasingly, responses of genes involved in sex differentiation, e.g., Anti-Müllerian Hormone (*amh*) are assisting the monitoring of the early effects of hormone treatments on treated fish [32]. The *amh* encodes a glycoprotein known as Müllerian inhibiting substance, a member of the transforming growth factor β superfamily of growth and differentiation factors [33]. In mammals, *amh* produced by Sertoli cells in fetal testes causes the regression of the Müllerian ducts, which in females differentiate into the Fallopian tubes and uterus and inhibits the expression of aromatase (*Cyp19a1*), the enzyme that converts androgens to estrogens [34]. In teleost fish, although Müllerian ducts are absent, the expression of *amh* occurs predominantly in Sertoli cells surrounding the spermatogonia [35], whose activation serves as an early and effective indicator of masculinization [32]. 

There is no study demonstrating masculinization in *G. holbrooki*, with the exception of the masculinizing effects of environmental pollutants in which females living in effluent waters display secondary sexual characters, e.g., gonopodium-like structures [22,36,37,38]. Such changes to sexual characteristics were thought to be caused by androgenic stimulation of the effluent water rather than perturbations to steroidogenic pathways in fish [7] with an exception of paradoxical masculinization caused by diethylstilbesterol (DES), a feminizing hormone, treatment of neonates [39]. 

In one of our earlier experiments at delivering MT to embryos via gravid females, it triggered the abortion of embryos and the treated females appeared to be masculinized by the hormone treatment. Based on these observations and the susceptibility of fish in general to exogenous androgens, we hypothesized that a ‘gonochorist’ such as *G. holbrooki* could retain an ability to recapitulate protogyny (similar to sequential hermaphrodites) in adulthood and thus provide an ideal system to investigate the mechanisms of hermaphroditism. In practice, the ability to sex-reverse adults (i.e., animals of known sex) could circumvent the need to treat larval stages that require drawn-out rearing and progeny testing to identify sex-reversed individuals for both aquaculture [40] and pest management [39] applications. Specifically, hormonal sex reversal has emerged as central to developing genetic strategies such as the Trojan Y [41] for controlling this notorious pest fish [42,43]. To test these possibilities, the effects of MT oral delivery on gonopodial development, gonad morphology, behavior and *amh* gene expression were investigated in maiden gravid (MG) and repeat gravid (RG) females.

## 2. Materials and Methods

### 2.1. Wild Specimens

Ten mature males (*Gambusia holbrooki*) collected in the Tamar Island Wetlands Reserve (TIWR) were used to characterize the morphology of mature male gonopodia and gonads. Specifically, fish (*n* = 5) were anesthetized and their anal fins photographed. For histological observation, 5 separate males were euthanized and fixed in neutral buffered formalin (10%). All animal experiments including collection, handling and transportation were conducted as stipulated by the Inland Fisheries Service Tasmania (Permit No. 2017-53) and approved by the University of Tasmania Animal Ethics Committee (Permit No. A0015354).

### 2.2. Experimental Fish and Rearing Conditions

Masculinization of two types of females were tested: those that had previous history of parturition (i.e., RG females; 38–53 mm in total length) and those just entering breeding season (MG females; 25–33 mm in total length). The RG females were collected from the wild and maintained under laboratory conditions and had a prior history of parturition before testing. The MG females, i.e., small-sized females (25–33 mm) with no prior parturition history were also collected from the wild and held in 50 L tanks to acclimatize to laboratory conditions for a week before allocating to the experiment. Typically, early in the breeding season, the overwintering and RG females (>38 mm) can be easily distinguished from young MG females by their relative size. The females were chosen randomly from acclimatization tanks for the study. Each female was anesthetized to measure weight, length and their anal fin photographed before being transferred to and individually held in a 2 L static tank (0 ppt, 25 °C, 16 L: 8 D-light turned on at 06.00 h).

MT was incorporated in commercial fish pellets (TetraMin^®^ tropical granules, Melle, Germany) and its effects tested on both groups of females. In experiment 1, RG females were fed twice daily to satiation one of two control diets (C1: normal feed with no chemical exposure or C2: normal feed mixed with 70% ethanol as vehicle control) or one of five treatments of MT enriched feed at the doses of 50, 75, 100, 150 and 200 mg/kg diet (5 females per treatment; Total = 35). In experiment 2, MG females were used with identical experimental design as experiment 1, except control C2 was excluded to increase replicates of experiment 2 (*n* = 12 per treatment; Total = 72) as it had no effect on morphological masculinization of the treated females in experiment 1. Every seven days, the treated females were anesthetized and their anal fins photographed. The treatments lasted for 50 days when the treated fish were anesthetized and their total length and weight measured. All treated females (except treated MG females in the 50 mg MT/kg diet group) were euthanized and sampled for histology, *amh* expression while the MG females treated with MT at the dose of 50 mg/kg diet were used to observe their behavior to normal virgin females. Water quality of the tanks was maintained at an optimum range by batch exchanging water every three days using temperature acclimated and conditioned freshwater.

### 2.3. Condition Factor

Condition factor (*K*) was calculated as an indicator of general ‘well-being’ of the fish, e.g., [44] as well as male and female body shapes. The females in each treatment group were measured for weight and length before and after the treatments occurred. The Fulton’s *K* factor of each female before and after treatments was calculated as described by Rätz and Lloret [45].
$$K=100\left(\frac{W}{L^{3}}\right)$$
where *W* is the weight (g) and *L* total length (cm). The mean condition factor for each treatment group was determined from the average *K*.

### 2.4. Photography and IMAGE Analysis

The male gonopodia and treated female anal fins were measured following the protocol described in [20]. Prior to measuring, all images were calibrated from pixels to micron using a stage micrometer at the respective magnifications. The calibrated images were used to measure lengths of anal fin rays 4 and 6 and widths of anal fin rays 3 and 4, using Fiji software [46]. Since anal fin ray 6 does not elongate during the formation of the gonopodium, the ratio of the lengths of anal fin rays 4 and 6 was calculated and used as a relative index of elongation. Normally, anal fin rays 3 and 4 are approximately the same thickness in immature fish and females but not in males. Therefore, the ratio of the widths of anal fin rays 3 and 4 was calculated and used as the degree of thickening of anal fin ray 3. The number of serrae and hooks were counted, and mean numbers were calculated in treated females and normal males.

### 2.5. Gonad Histology

As the primary objective was to test functional masculinization by mating with an untreated virgin female, only one female per treatment (including controls) was randomly sampled for histological examination. In addition, five control males were sampled to serve as controls.

The sampled fish were euthanized using an overdose of AQUI-S (400 mg/L) followed by decapitation. The caudal fin of each fish was trimmed in preparation for histological processing. The trimmed abdominal sections were fixed in neutral buffered formalin (10%). The histological techniques of sectioning and staining for all samples were as described before [47]. After fixing, the samples were further trimmed and placed in cassettes for processing. The cassettes were processed through a graded ethanol series for dehydration, cleared with xylene, embedded in paraffin before being longitudinally sectioned (3 or 5 µm). To standardize for gonad area measurements, median sagittal sections were made by trimming the paraffin block until the spine was just exposed and mounted on glass slides. After staining with haematoxylin and eosin, the slides were observed under a compound microscope and micro-photographed. From the micro-photographs, the gonad area of each treated female was measured by using Fiji software [46]. When testicular tissue was present in the gonads of treated females, the area was measured and percentage of testicular tissue area was calculated. In addition, the gonad histology of the treated females, control females and males was compared descriptively. 

### 2.6. Gene Expression

The effects of MT treatments on the expression of *amh* were observed using quantitative PCR (qPCR). Immediately after termination of the treatments, treated fish were euthanized by using an overdose of AQUI-S (400 mg/L). In experiment 1 (MT treated RG females), 21 treated females (*n* = 3 fish per treatment and the two controls) were sampled, while in experiment 2 (MT treated MG females) 15 treated females (*n* = 3 fish per treatment and the control) were sampled. The sampled fish were dissected, and the gonads were individually collected, fixed in RNA*later*^TM^ (Thermo Fisher Scientific, Scoresby, VIC, Australia) and stored at minus 80 °C until extraction for RNA.

#### 2.6.1. Total RNA Extraction, gDNA Removal and Reverse Transcription

Total RNA extraction, gDNA removal and reverse transcription were conducted as described previously [32]. Total RNA was extracted from the gonads using the RNeasy Plus Mini Kit (Qiagen, Chadstone, VIC, Australia). Tissues were lysed by repetitive aspirations through a sterile 19 and 25-gauge needle connected to a 3 mL sterile syringe, to produce a homogenized lysate. Column removal of gDNA and subsequent purification of lysate followed those as described in the manufacturer’s protocols. RNA integrity was examined for the 18 and 28 S rRNA bands using gel electrophoresis through a 1% agarose in Tris-acetate-EDTA (TAE) buffer. The concentration of total RNA was determined using Qubit RNA HS Assay Kit (Life Technologies, Mulgrave, VIC, Australia). A total of 60 ng uL^−1^ of total RNA was used for reverse transcription using the Tetro Reverse Transcriptase and Ribosafe RNase inhibitor (Bioline, Eveleigh, NSW, Australia) as per the manufacturer’s protocols. The cDNA was stored at minus 20 °C and used without dilution in quantitative PCR.

#### 2.6.2. Gene Expression Analysis

Quantitative PCR was performed using a CFX96 Touch real time PCR detection system (Bio-Rad, Gladesville, Australia). The primers for target *amh* were: ghAmhF644 5′-CCCCTGCAGATGGAGAGCTGGGCGTCATTT-3′ and ghAmhR553 5′-AACGTCGTCCCTGAARTGCAAGCAGA -3′ which yielded a PCR amplicon of 88 bp. The primers for the reference *GAPDH*: ghGAPDHF2 5′-AGCCAAGGCTGTTGGCAAGGTCATC-3′ and ghGAPDHcDNAR2 5′-GTCATCATACTTGGCTGGTTTCTCC-3′ yielded a 133 bp amplicon [32]. The quantitative PCR was carried out in 10 µL volume containing 5 µL of 2 × iTaq Universal SYBR Green Supermix (Bio-Rad, Gladesville, NSW, Australia), each primer at final 0.4 µM, and 1 µL of cDNA. Triplicate tubes were used for each sample. Thermal cycling conditions encompassed the initial denaturation at 95 °C for 1 min and 40 amplification cycles involving 95 °C for 5 s, 64 °C for 5 s and 72 °C for 5 s. A melt curve analysis was used which included initial denaturation at 95 °C for 10 s, followed by 0.5 °C increment every 5 s from 72 °C to 90 °C. Melt curve analysis consistently showed single peak product corresponding to the targeted *amh* and *GAPDH* amplicon as was also confirmed by gel electrophoresis. Gene expression analysis was carried out using Bio-Rad CFX Manager 3.1 using default settings. Briefly, the cycle of quantification (Cq) was determined using the mode regression, and expression was normalized to double delta Cq (∆∆Cq).

### 2.7. Behavioural Interactions

A behavioral experiment was conducted immediately (i.e., within 2 days) after the MT treatment ceased. Limited number of cameras prevented simultaneous recording of all fish. The females treated with 50 mg MT/kg diet (Experiment 2; MG females), showing the best gonopodial development (test fish), were used to observe how these females interact with normal mature females (stimulus fish). There were three groups of fish: group 1 (control males, *n* = 5), group 2 (control females, *n* = 11) and group 3 (50 mg MT/kg treated females, *n* = 11) paired with virgin females. The experimental fish were held individually before the experiment was conducted. To avoid aggressive behavior, i.e., nipping caused by hunger, the test fish and stimulus fish were fed for 15 min before being transferred to a test tank (28 × 14 × 10 cm) set up with a black background and an in-front camera (GroPro, version 3) located 6 cm away from the test tank. The trial was recorded for 30 min. The time frame of 30 min was chosen based on previous studies [37,48,49]. The recorded footage was analyzed on play back to observe behavioral interactions of the test fish (i) mating behavior, i.e., the number of mating attempts defined as gonopodial thrusts made after being initially positioned below and slightly behind females [49] and (ii) aggressive behavior, i.e., the number of nips after quickly approaching.

### 2.8. Statistical Analysis

The data were analyzed using IBM SPSS Statistic software (version 24). Normality (Shapiro–Wilk test) and homogeneity of variances (Levene’s test) were examined prior to applying ANOVA analyses. One-way ANOVA analysis followed by Tukey HSD or Games-Howell post hoc test (where applicable) was used to observe differences in condition factors of the females in treatment groups before and after feeding the MT-enriched diet and determine the statistical significance of *amh* expression changes. Two-way mixed ANOVA [50] was used to perform mixed effects model analysis of the effects of MT treatment doses on length ratios of fin rays 4 and 6 and width ratios of fin rays 3 and 4 over 8 time points during a 50-day treatment period. The treatment groups and hormone exposure time (with interaction term) were entered as fixed effects into the model. Prior to two-way mixed ANOVA, the data were tested for outliers, homogeneity of variances, homogeneity of covariances as assessed by the examination of studentized residuals for values greater than ±3, Levene’s test of homogeneity of variance, Box’s test of equality of covariance matrices, respectively. Additionally, the assumption of sphericity was assessed by Mauchly’s test of sphericity and Greenhouse–Geisser correction was used if the assumption of sphericity was violated. Kruskal–Wallis H tests [51] were conducted to determine if there were differences in the number of serrae and hooks between the treatment and control groups at day 50 as well as the number of mating attempts between the three fish groups (control males, control females and MT treated females) paired with virgin females. Distributions of the number of serrae, hooks and mating attempts were assessed by visual inspection of a boxplot. Pairwise comparisons were performed using Dunn’s (1964) as cited in Laerd Statistics [51] procedure with a Bonferroni correction for multiple comparisons. Adjusted *p*-value was presented. In all analyses, differences were significant at *p* < 0.05.

## 3. Results

### 3.1. Gonopodial and Body Morphology of Adult Females Following MT Treatment

In females treated with MT (i.e., both groups), the anal fin rays (except the control females) started to elongate and the gravid spots faded between 9 and 11 days post-exposure to MT. At 30 days post-exposure, the treated females had not yet fully developed the gonopodium-like structures. At the end of the 50-day treatment period, treated females in experiment 2 (MG females) developed complete gonopodium-like structures, with the exception of treated females in 200 mg MT/kg diet group. Anal fin rays 3 of these females were thickened and the anal fin rays 3, 4 and 5 of the same females were elongated and formed a gonopodial tip equipped with hooks and serrae (Figure 1E,F). During the treatment period, the treated females also exhibited slender bellies similar to those of normal males (Figure 1G,I). One week after termination of the treatment, their bellies began to swell, but the gonopodia remained and they did not develop a gravid spot. Detailed results for both RG and MG female groups are presented separately below.

#### 3.1.1. Repeat Gravid Females Treated with MT

At the end of the 50-day treatment period, the RG females treated with MT in all treatments developed short gonopodium-like structures with incomplete gonopodial tips, where anal fin ray 3 did not elongate adequately to combine with anal fin rays 4 and 5 to form fully developed complex terminal structures (Figure 1C,D) in any treated females. 

There was a significant difference in the mean final condition factors between RG females fed control diets and MT-enriched feed (one-way ANOVA, *F*_(6,24)_ = 7.43, *p* < 0.0005), although the mean initial condition factors of these females were not significantly different (one-way ANOVA, *F*_(6,9.9)_ = 1.22, *p* = 0.37). Specifically, the mean final condition factors of the treatment groups of 50, 100 and 200 mg MT/kg diet were significantly smaller than those of the control groups (Figure 2a).

There was evidence of an interaction effect between the hormone treatment doses and exposure duration on the ratio of the lengths of fin rays 4 and 6, *F*_(31.1,124.3)_ = 8.05, *p* < 0.0005, ƞ^2^ = 0.67 (Figure 3a). There was also a statistically significant effect of time on the ratio of the lengths of fin rays 4 and 6 for 50 mg/kg group (*F*_(7,21)_ = 16.78, *p* < 0.0005, ƞ^2^ = 0.85) and 75 mg/kg group (*F*_(7,28)_ = 28.55, *p* < 0.0005, ƞ^2^ = 0.88). The results of pairwise comparisons showed that the ratio of the lengths of fin rays 4 and 6 of the females exposed to 75 and 50 mg MT/kg rose significantly and reached a peak at 14- and 21-day exposure, respectively, before remaining stable during the rest of the 50-day treatment period, while there was no statistically significant increase in the ratios of the lengths of fin rays 4 and 6 of the females in the 100, 150, 200 mg MT/kg diet and control groups through the treatment period. At 21-day exposure, there was a statistically significant difference in the ratio of the lengths of fin rays 4 and 6 between MT-exposed groups, *F*_(7,28)_ = 332.24, *p* < 0.0005, ƞ^2^ = 0.99. The length ratio was significantly greater in the 50 mg/kg group (1.34 ± 0.03) compared to those of females in control 1 (1.05 ± 0.03, *p* < 0.0005), control 2 (1.08 ± 0.03, *p* < 0.0005), 200 mg/kg (1.1 ± 0.04, *p* < 0.0005), 150 mg/kg (1.12 ± 0.03, *p* < 0.0005), 100 mg/kg (1.22 ± 0.03, *p* = 0.001) and 75 mg/kg (1.26 ± 0.03, *p* = 0.03) group, but significantly smaller than that of the control male group (2.32 ± 0.03, *p* < 0.0005).

There was a statistically significant interaction between MT treatment doses and time on the ratio of the widths of fin rays 3 and 4, *F*_(16.4,196)_ = 6.4, *p* < 0.0005, ƞ^2^ = 0.62 (Figure 3b). There was a statistically significant effect of time on the ratio of the widths of fin rays 3 and 4 for 75 mg/kg group (*F*_(7,28)_ = 16.25, *p* < 0.0005, ƞ^2^ = 0.8) and 100 mg/kg group (*F*_(7,21)_ = 26.94, *p* < 0.0005, ƞ^2^ = 0.9). The results of pairwise comparisons indicated that the ratio of the widths of fin rays 3 and 4 of the females exposed to 75 and 100 mg MT/kg diet increased and peaked at 28-day exposure before remaining stable during the rest of the 50-day treatment period, while there was no significant increase in the ratios of the widths of fin rays 3 and 4 of the females in the 50, 150, 200 and control groups during the treatment period. At 28-day exposure, there was a statistically significant difference in the ratio of the widths of fin rays 3 and 4 between MT-exposed and control groups, *F*_(7,28)_ = 34.6, *p* < 0.0005, ƞ^2^ = 0.9. The ratio of the widths of fin rays 3 and 4 of 75 mg/kg group (1.5 ± 0.13) was significantly greater than those of the control 2 group (1.0 ± 0.12, *p* = 0.001), and the control 1 group (1.01 ± 0.12, *p* = 0.001), but significantly smaller than that of the control male group (2.73 ± 0.13, *p* < 0.0005). This width ratio was not significantly different from those of 50 mg/kg (1.36 ± 0.14, *p* = 0.32), 150 mg/kg (1.44 ± 0.13, *p* = 0.66), 100 mg/kg (1.48 ± 0.14, *p* = 0.91) and 200 mg/kg (1.5 ± 0.15, *p* = 1.0) groups. 

There was a statistically significant interaction between MT treatment doses and time on the ratio of the widths of fin rays 3 and 4, *F*_(16.4,196)_ = 6.4, *p* < 0.0005, ƞ^2^ = 0.62 (Figure 3b). There was a statistically significant effect of time on the ratio of the widths of fin rays 3 and 4 for 75 mg/kg group (*F*_(7,28)_ = 16.25, *p* < 0.0005, ƞ^2^ = 0.8) and 100 mg/kg group (*F*_(7,21)_ = 26.94, *p* < 0.0005, ƞ^2^ = 0.9). The results of pairwise comparisons indicated that the ratio of the widths of fin rays 3 and 4 of the females exposed to 75 and 100 mg MT/kg diet increased and peaked at 28-day exposure before remaining stable during the rest of the 50-day treatment period, while there was no significant increase in the ratios of the widths of fin rays 3 and 4 of the females in the 50, 150, 200 and control groups during the treatment period. At 28-day exposure, there was a statistically significant difference in the ratio of the widths of fin rays 3 and 4 between MT-exposed and control groups, *F*_(7,28)_ = 34.6, *p* < 0.0005, ƞ^2^ = 0.9. The ratio of the widths of fin rays 3 and 4 of 75 mg/kg group (1.5 ± 0.13) was significantly greater than those of the control 2 group (1.0 ± 0.12, *p* = 0.001), and the control 1 group (1.01 ± 0.12, *p* = 0.001), but significantly smaller than that of the control male group (2.73 ± 0.13, *p* < 0.0005). This width ratio was not significantly different from those of 50 mg/kg (1.36 ± 0.14, *p* = 0.32), 150 mg/kg (1.44 ± 0.13, *p* = 0.66), 100 mg/kg (1.48 ± 0.14, *p =* 0.91) and 200 mg/kg (1.5 ± 0.15, *p* = 1.0) groups. 

There was a statistically significant interaction between MT treatment doses and time on the ratio of the widths of fin rays 3 and 4, *F*_(16.4,196)_ = 6.4, *p* < 0.0005, ƞ^2^ = 0.62 (Figure 3b). There was a statistically significant effect of time on the ratio of the widths of fin rays 3 and 4 for 75 mg/kg group (*F*_(7,28)_ = 16.25, *p* < 0.0005, ƞ^2^ = 0.8) and 100 mg/kg group (*F*_(7,21)_ = 26.94, *p* < 0.0005, ƞ^2^ = 0.9). The results of pairwise comparisons indicated that the ratio of the widths of fin rays 3 and 4 of the females exposed to 75 and 100 mg MT/kg diet increased and peaked at 28-day exposure before remaining stable during the rest of the 50-day treatment period, while there was no significant increase in the ratios of the widths of fin rays 3 and 4 of the females in the 50, 150, 200 and control groups during the treatment period. At 28-day exposure, there was a statistically significant difference in the ratio of the widths of fin rays 3 and 4 between MT-exposed and control groups, *F*_(7,28)_ = 34.6, *p* < 0.0005, ƞ^2^ = 0.9. The ratio of the widths of fin rays 3 and 4 of 75 mg/kg group (1.5 ± 0.13) was significantly greater than those of the control 2 group (1.0 ± 0.12, *p* = 0.001), and the control 1 group (1.01 ± 0.12, *p* = 0.001), but significantly smaller than that of the control male group (2.73 ± 0.13, *p* < 0.0005). This width ratio was not significantly different from those of 50 mg/kg (1.36 ± 0.14, *p* = 0.32), 150 mg/kg (1.44 ± 0.13, *p* = 0.66), 100 mg/kg (1.48 ± 0.14, *p =* 0.91) and 200 mg/kg (1.5 ± 0.15, *p* = 1.0) groups. 

Although all RG females treated with MT showed signs of anal fin elongation at the end of the 50-day treatment period, none of them completely developed complex terminal structures such as serrae and hooks. Fin ray 3 did not elongate adequately to combine with fin rays 4 and 5 to form a gonopodial tip.

#### 3.1.2. Maiden Gravid Females Treated with MT

At the end of the 50-day treatment period, MG females treated with MT showed better gonopodial development than the RG females. In fact, most of the females developed longer gonopodium-like structures with complete gonopodial tips equipped with serrae and hooks (Figure 1E,F).

In terms of condition factor, the mean initial condition factors of the MG females were not significantly different (*F*_(5,64)_ = 0.85, *p* = 0.52), but there was a significant difference in the mean final condition factors of the females in the treatment and control groups (*F*_(5,64)_ = 15.09, *p* < 0.0005). Follow-up Tukey HSD tests showed that the females in the control group had the highest mean condition factor which was significantly different from those of the treatment groups (Figure 2b).

There was a statistically significant interaction between the MT treatment doses and time on the ratio of the lengths of fin rays 4 and 6, *F*_(28.9,337.7)_ = 42.21, *p* < 0.0005, ƞ^2^ = 0.78 (Figure 4a). There was a statistically significant effect of time on the ratio of the lengths of fin rays 4 and 6 for 50 mg/kg group (*F*_(7,70)_ = 123.95, *p* < 0.0005, ƞ^2^ = 0.93), 75 mg/kg group (*F*_(7,70)_ = 166.98, *p* < 0.0005, ƞ^2^ = 0.94), 100 mg/kg group (*F*_(7,70)_ = 100.59, *p* < 0.0005, ƞ^2^ = 0.91), 150 mg/kg group (*F*_(7,70)_ = 109.9, *p* < 0.0005, ƞ^2^ = 0.92) and 200 mg/kg (*F*_(7,70)_ = 59.53, *p* < 0.0005, ƞ^2^ = 0.86). The results of pairwise comparisons showed that the ratio of the lengths of fin rays 4 and 6 of the females exposed to MT rose significantly after 7-day exposure to MT and peaked at 14-day exposure (except the 50 mg/kg group gaining a peak at 21-day exposure) before remaining stable during the rest of the 50-day treatment period, while those of the controls remained unchanged for the whole period (Figure 4a). At 21-day exposure, there was a significant difference in the ratio of the lengths of fin rays 4 and 6 between MT-exposed groups, *F*_(6,70)_ = 136.42, *p* < 0.0005, ƞ^2^ = 0.92. The length ratio was significantly greater in the 50 mg/kg group (1.82 ± 0.38) compared to those of females in the control group (1.33 ± 0.38, *p <* 0.0005), 150 mg/kg group (1.70 ± 0.38, *p* = 0.23) and 200 mg/kg group (1.64 ± 0.38, *p* < 0.0005), but significantly smaller than that of the male control group (2.37 ± 0.38, *p* < 0.0005). The length ratios of fin rays 4 and 6 of the 75 mg/kg group (1.78 ± 0.38, *p* < 0.0005) and the 100 mg/kg group (1.77 ± 0.38, *p* < 0.0005) were statistically significantly greater than that of the female control group, although these ratios were not significantly different from those of 50, 150 and 200 mg/kg groups. 

There was statistically significant interaction between MT treatment doses and time on the ratio of the widths of fin ray 3 and 4, *F*_(24,490)_ = 7.23, *p* < 0.0005, ƞ^2^ = 0.38 (Figure 4b). There was a statistically significant effect of time on the ratio of the widths of fin rays 3 and 4 for 50 mg/kg group (*F*_(3,30)_ = 15.52, *p* < 0.0005, ƞ^2^ = 0.61), 75 mg/kg group (*F*_(7,70)_ = 20.38, *p* < 0.0005, ƞ^2^ = 0.67), 100 mg/kg group (*F*_(2.2,22)_ = 20.32, *p* < 0.0005, ƞ^2^ = 0.67), 150 mg/kg group (*F*_(2.8, 27.8)_ = 34.61, *p* < 0.0005, ƞ^2^ = 0.78) and 200 mg/kg group (*F*_(2.7,27)_ = 44.3, *p* < 0.0005, ƞ^2^ = 0.81). The results of pairwise comparisons indicated that the ratio of the widths of fin rays 3 and 4 of the females exposed to 50, 75, 100, 150 and 200 mg MT/kg diet increased significantly and gained peaks at 28-day exposure before remaining stable during the rest of the 50-day treatment period, whereas that ratio of the females in control groups remained unchanged during the treatment period. At 28-day exposure, there was a statistically significant difference in the ratio of the widths of fin rays 3 and 4 between treatment groups, *F*_(6,70)_ = 23.9, *p* < 0.0005, ƞ^2^ = 0.67. The ratio of the widths of fin rays 3 and 4 of 75 mg/kg group (2.09 ± 0.08) was greater than that of the control female group (1.65 ± 0.08, *p* < 0.0005), but was significantly smaller than that of the control male group (2.6 ± 0.08, *p* < 0.0005). This ratio was not significantly different from those of 50 mg/kg group (2.02 ± 0.08, *p* = 0.97), 200 mg/kg group (2.02 ± 0.08, *p* = 0.98), 100 mg/kg group (2.08 ± 0.08, *p* = 1.0) and 150 mg/kg group (2.09 ± 0.08, *p* = 1.0).

In contrast to RG females, most of the MG females treated with MT elongated their anal fins and completely developed gonopodium-like structures with gonopodial tips equipped with serrae and hooks, while the females in the control group maintained their round shapes of anal fins. The serrae and hooks were seen from day 21 of the treatment period. Median numbers of serrae were significantly different between treatment groups (Kruskal–Wallis H test, χ^2^_(6)_ = 54.07, *p* < 0.0005, *n* = 11 for each group). Importantly, pairwise comparisons analysis revealed that there was no statistically significant difference in median numbers of serrae between 50 mg/kg group (5 serrae, *p* = 0.66), 100 mg/kg group (5 serrae) (*p* = 0.06) and control male group (7 serrae), but these median numbers were significantly different from the control female group (0 serrae) with *p* < 0.0005 and *p* = 0.002, respectively. 

There were significant differences in median numbers of hooks between treatment groups (Kruskal–Wallis H test, χ^2^_(6)_ = 69.18, *p* < 0.0005, *n* = 11 for each group). Pairwise comparisons indicated that all MT-exposed groups attained two hooks which was significantly different from the control female group (no hook, *p* < 0.0005), but not significantly different from the control male group (2 hooks, *p* = 1.0). There was no significant difference in the median numbers of hooks among MT-exposed groups.

### 3.2. Gonad Histology

#### 3.2.1. Control Female and Male Gonads

Control females typically had ovaries filled with mature oocytes at the vitellogenic stage (Figure 5A,B), while control males possessed a funnel-shaped testis which was distinguishable into two parts: an anterior glandular part (testis lobe) and a posterior muscular vas deferens (Figure 5C). The glandular part could be further distinguished into a network of efferent ducts located medially which was surrounded by a lateral layer of secretory testicular tissue (Figure 5C,D).

#### 3.2.2. Repeat Gravid Females Treated with MT

After a 50-day exposure to MT, the gonads of the RG females in treatment groups were relatively shrunken, as was also evident by the reduced gonad area (Table 1), except the female in the 50 mg MT/kg diet despite showing signs of follicular atresia in histological sections. Primary spermatocytes (S1) were observed in the ovaries of most treated females (Figure 6 and Figure 7), except the female in the 100 mg MT/kg group. The female treated with 200 mg MT/kg possessed the largest area of testicular tissue (8.2%) where there were several stages of spermatocyte development throughout the ovary including primary spermatocytes (S1), secondary spermatocytes (S2), early (S3) and late (S4) stages of spermatids (Figure 7). 

#### 3.2.3. Maiden Gravid Females Treated with MT

After a 50-day exposure to MT, the gonads of the MG females in the treatment groups were shrunken (Table 2). In contrast to the results of RG females treated with MT, there was no testicular tissue in the ovaries of MG females treated with MT at doses 75, 100, 150 and 200 mg/kg. However, the MG female in the treatment group of 50 mg/kg was stimulated to develop spermatogenic tissue (45% of the total gonad area) with all stages of spermatocyte development (Figure 8). Additionally, Sertoli cells were observed in the gonad of MG females treated with the 50 mg MT/kg diet (Figure 9). Interestingly, the testis tissue had a similar shaped contour as that of the control male, with anterior glandular tissue and signs of posterior vas deferens. However, internally the tissue was less organized compared to control males.

### 3.3. Expression of Amh in Ovaries of Treated Fish

There was no significant difference between corresponding treatment groups (doses) between MG- and RG-treated females (Figure 10). However, the gonads of females fed with the 50 mg MT/kg diet showed a significant upregulation of *amh* expression in both RG (mean of 9-fold increase, SE ± 0.48, *p* < 0.0005,) and MG (mean of 6-fold increase, SE ± 0.47, *p* < 0.005) compared to respective control fish. Interestingly, higher doses of MT from 100 to 200 mg/kg at 50 mg intervals did not elicit a response; instead, they were comparable (*p* > 0.05) to those of the control. There was also no notable decrease in the expression with an increase in the dosage of MT. 

### 3.4. Behavioral Interactions

Median numbers of mating attempts were significantly different between fish groups (Kruskal–Wallis H test, χ^2^_(2)_ = 9.14, *p* = 0.01). Post hoc test revealed significant differences in median numbers of mating attempts between the MT-treated female group (10 mating attempts) and control female group (0 mating attempt, *p* = 0.029); control male group (23 mating attempts) and control female group (0 mating attempt, *p* = 0.016), but not between MT-treated female group and control male group (*p* = 1.0). No aggressive behavior between treated females/control females/control males and virgin females was recorded during a 30 min period.

### 3.5. Stability of Sex Reversal

Before exposure to MT, 80 percent of females chosen for the experiments possessed a gravid spot (Table 3). However, after being treated with MT, all exposed fish lost their female identity as their gravid spots faded but presented male characteristics, e.g., gonopodia. Post-treatment cessation, there was only 1.7% of the treated females reverting back to their original sex and giving birth to young.

## 4. Discussion

MT was found to elicit multiple masculinizing effects on the treated females including secondary sexual characters (anal fin and body morphology), gonad morphology, expression of *amh* and male mating behavior. Collectively, the MT exposure appears to trigger a sex change process of adult females to males in the gonochorist *G. holbrooki,* similar to what occurs naturally in hermaphroditic fish (protogyny), e.g., as in Red Sea fish, *Anthias squamipinnis* where females change sex to males [52], suggesting the sex can be plastic even in adulthood in this gonochoristic fish. 

### 4.1. MT Exposure Had a Masculinizing Effect on Anal Fin, Body Shape and Gonad Morphology

MT dietary administration of all concentrations stimulated the development of gonopodial-like structures of the anal fins in females exposed to MT, nearly mimicking those of normal males in some instances. This is similar to the observations in *G. affinis* [20,53], a sister species to *G. holbrooki* and was particularly prominent in MG compared to RG females, suggesting that the degree of secondary sexual character masculinization was influenced by the age of the females or their parturition history. RG females were more than three years old, while MG females were predicted to be less than one year old as *G. holbrooki* rarely live longer in the wild at the study site [54]. The older age may have caused a decline in hormone receptors and metabolic activity [55,56], conceivably making the animals less responsive to hormone treatment. However, such physiological changes associated with repeat parturition cannot be completely ruled out.

Not surprisingly, the exposure of fish to MT had a significant loss in condition of the female fish. This loss in condition is attributed to the shrinking of ovaries in readiness for re-modelling to testes and male body shapes that are generally slender and lighter than females in this species. Indeed, the treated females showed slender bellies which was consistent with the results of gross examination that the gonads of the treated females had shrunk. These observations are comparable to atrophied gonads observed in ET- and DES-treated catfish [57] and DES-treated *G. holbrooki* [58]. Alternatively, the decreased condition factors could reflect general loss in ‘well-being’ of fish, caused by reduced food intake and increased metabolic expenditure for detoxification and maintenance of the normal body functions [44]. However, this is unlikely as animals fed normally and did not succumb to adverse mortalities.

Besides transforming the anal fin to a gonopodium, there were parallel changes of the gonad of the treated females to those resembling the testes. Specifically, the shrunken gonads of MT-treated females, suggested ovarian atrophy, which was accompanied by the appearance of testicular tissue in the ovaries of treated females. Ovarian atrophy with interspersed testicular tissue is a common feature of hermaphroditic fish undergoing sex change [59]. Collectively, these observations validated masculinization of the ovaries, suggesting bipotentiality of the adult gonads in this species. However, the age/reproductive status influenced the extent of gonad maculinization as a relatively low dose of MT was required to stimulate gonad masculinization of MG females, compared to RG females. This agrees with the results of the secondary sexual character development in treated MG females, where the two lowest doses stimulated best gonopodial development with complete gonopodial tips, unlike RG females which did not develop complete gonopodial tips. The formation of testicular tissue was further confirmed by the presence of Sertoli cells and importantly spermatids were organized in rosettes, facilitating the formation of spermatozeugmata. Sertoli cells are necessary for the formation of spermatozeugmata and synthesis of male steroid hormones in poeciliids [60]. Although the development of secondary sexual characters has been reported in adult female mosquitofish exposed to androgens [20,22,61], this is the first report showing spermatocyte development inside the ovaries (i.e., ovotestis) of the female mosquitofish treated with MT during the adult stage. A similar observation was reported in female medaka (*Oryzias latipes*) exposed to 5 ng/L of TB [29] but during early sex differentiation (from 15–25 dpf) rather than the adult stage. Typically, in sex-reversal studies, hormone treatment is applied immediately post-hatching and/or juvenile stages to masculinize fish [9]. In contrast, this study demonstrates a masculinizing effect of MT on adult fish where MT triggered sex reversal in *G. holbrooki*, similar to those that occur in hermaphrodites (protogyny). 

The limited number of samples subjected to histological examinations may have precluded the detection of complete anatomical transformation of the ovary to testis as the responses of individual fish within treatment groups can vary. This is likely as individual variations between fish among treatments were evident in gonopodial development. Such differences in response to MT treatment among treated females could be attributed to individual variation in metabolic rates [62] or reproductive status that are known to occur in this and many fish species [63]. Alternatively, despite producing male gametes, there might remain some anatomical hurdles for sex-reversed individuals to express gametes and contribute to reproduction naturally. A relatively less structured testicular tissue and indistinct lumens in the sex-reversed fish may suggest this possibility. Such limitations to MT sex-reversed salmonids are known to occur, which require artificial propagation with more recent suggestions that these can be overcome by employing appropriate modes or doses of hormone administration [40]. In this regard, the findings of this study may speed up the production of sex-reversed fish, which could assist downstream investigations on sex differentiation/determination as well as production of monosex populations for controlling pest population of this species. In the latter application, a harvest of wild adults for sex reversal serves the dual purpose of contributing to the immediate suppression of populations and as broodstock for raising genetic carriers for long-term genetic eradication strategies.

### 4.2. The Amh Response Suggests a Narrow Physiological Window of Susceptibility for Sex Reversal of Adult Females

The *amh* is known to have a primary role in male sex differentiation in this species [32]. Interestingly, the magnitude of the *amh* response in both 50 mg MT/kg diet treated RG (i.e., 9-fold) and MG (i.e., 6-fold) females were in broad agreement with a previous work that reported a similarly (6-fold) higher expression of *amh* in adult testis compared to ovarian tissue in this species [32]. This coincided with the prominent presence of testicular tissue in the animal treated at this dose, cross validating the results. Typically, *amh* is known to be produced by immature Sertoli cells and granulosa cells (to a lesser degree) from birth to the end of reproductive life [64]. It is likely that the ovaries of treated females (50 mg MT/kg diet) developed immature Sertoli cells, producing the upregulation of *amh*. This hypothesis is also supported by histological results in that Sertoli cells occurred more prominently in the ovaries of MG females treated with 50 mg MT/kg. Sertoli cells provide nutrients and a permeability barrier during spermatogenesis [65]. A lack of difference in *amh* expression between RG and MG females, particularly in the 50 mg/kg diet, is intriguing given marked differences were observed in gross morphological and histological responses in the groups. This may imply that RG females undergo significant post-parturition changes that restrict the ability of *amh to* suppress ovarian differentiation and hence their susceptibility to sex change. In practice, a significant *amh* response only at the 50 mg/kg diet treatment suggests that there exists a narrow physiological window at which the adult ovaries are susceptible to sex reversal. Conceivably, feedback mechanisms kick in to maintain the status quo outside this susceptible window. Collectively, the *amh* activation serves as a key indicator to refine treatment doses for sex reversal and for investigating global gene expression profiles to shed greater light on the cause and effects of the sex change process stimulated in this species and more broadly in protogynous hermaphrodites.

### 4.3. Treated Adult Females Mimic the Behavior of Normal Males

Male sexual behavior can be induced in female fish by exposure to androgens [66]. Consistent with this, the MG females treated with the 50 mg MT/kg diet showed mating behavior that was similar to normal males. This finding seems to parallel the observation in *G. affinis* [37] where the paper-mill effluent masculinized females exhibited behavioral patterns of typical males. Alteration of reproductive behavior in male *G. holbrooki* [67] paired with females exposed to 100 ng/L levonorgestrel, a human contraceptive progestin is also known to occur as treated females showed behavior atypical of normal females, preventing mating success. It is likely that the male sexual behavior in the treated females was driven by the hormone treatments rather than the fact that they were functionally sex reversed. However, the development of testis tissue as well as the activation of *amh* suggest that these are more likely the effect of stable sex reversal. Regardless, this needs further verification. 

### 4.4. Stability of Sex Reversal

Treated females were sex-reversed as their gravid spots were lost and they developed gonopodium-like structures as in normal males. Importantly, they presented continuing signs of veering the gonopodium-like structures from back to front in a similar manner to typical males during courtship with females. The ‘sex-reversed’ individuals maintained a male-like phenotype as none of them showed signs of regaining the gravid spots, with the exception of two. Specifically, one treated female parturiated 38 days post-treatment and the other at 57 days post-treatment cessation, without developing a gravid spot. It is likely that these two individuals, which had previously displayed male-like behavior, reverted back to their original sex when the effects of the exogenous androgen had faded. Alternatively, they had undergone sex reversal, but held the developing embryos (conceived before the onset of MT treatment) in a state of diapause owing to MT effect. The latter is likely as gonad histology of a MT-treated female, immediately at termination of treatment, had retained a fully grown embryo despite developing testicular tissue (Figure 8). Regardless, these observations suggest an ability of *G. holbrooki* adults to switch between sexes depending on their physiological status, an aspect that needs further investigation. There are no comparable observations in any other gonochoristic fish, but these are similar to those that occur in naturally sequential hermaphrodites.

## 5. Conclusions

Oral administration of MT to mature females indicated an ability to induce sex reversal, particularly at the lowest MT concentration (50 mg/kg diet) administered. This was able to stimulate male secondary sexual characteristics, upregulated expression of *amh* gene, the formation of testicular tissue as well as the alteration of normal behavior of MT-treated MG females (i.e., showing mating behavior). The hormone appeared to play a role as a trigger to switch sex from females to males similar to what happens naturally in protogynous fish. However, these effects may be incomplete posing anatomical hurdles for the animals to express milt and breed naturally or may reflect the need for further optimization. Regardless, this study provides strong evidence for plasticity of sex (even in adulthood) in this species which could be exploited to sex-reverse animals with known phenotypic sex, bidirectionally. Such an ability provides direct confirmation of sex reversal, circumventing the need for sex markers or long drawn-out progeny testing, expediting investigations of sex differentiation and determination as well as production of Trojan carriers [41,42] for controlling their pest populations.

## Figures and Tables

**Figure 1 biology-11-00694-f001:**
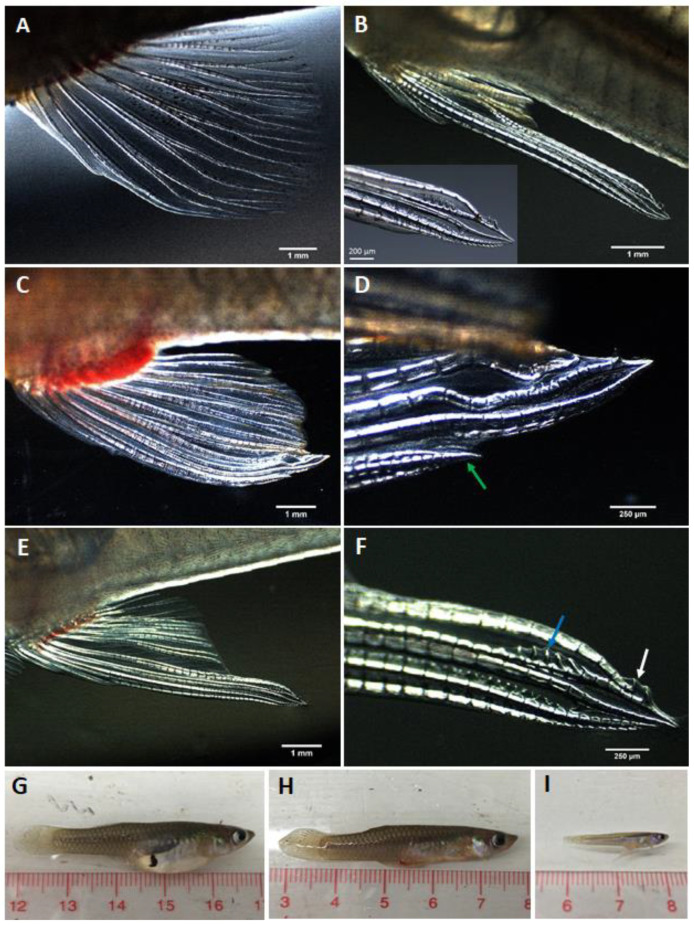
Shapes of anal fins and body shapes of control and MT-treated G. holbrooki. (**A**) The anal fin of a normal female, (**B**) and (**B**-inset) the gonopodium (modified anal fin) of a normal male, (**C**) the curved short gonopodium-like structures of RG females treated with 75 mg MT/kg diet post 50-day treatment period and (**D**) the gonopodial tip of the treated females (at higher magnification) indicating that fin ray 3 (green arrow) did not completely elongate to combine with fin rays 4 and 5 to generate fully developed complex terminal structures. (**E**) The gonopodium-like structure of MG females treated with 75 mg MT/kg diet post 50-day treatment period and (**F**) its tip (at higher magnification) equipped with serrae (blue arrow) and hooks (white arrow). Body shapes of a typical female (**G**), RG female treated with 150 mg MT/kg diet post 50-day treatment period (**H**) and a typical male (**I**).

**Figure 2 biology-11-00694-f002:**
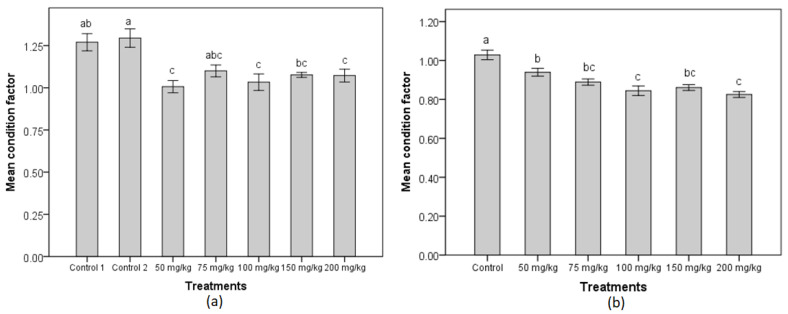
Condition factors of RG females and MG females post 50-day MT treatment period. (**a**) Mean final condition factors (±SE, *n* = 5) of RG females and (**b**) mean final condition (±SE, *n* = 12) of MG females. Means with a different letter are significantly different from one another. Control 1/control groups received feed without any hormone, while control 2 group received feed incorporated with 70% ethanol as vehicle control.

**Figure 3 biology-11-00694-f003:**
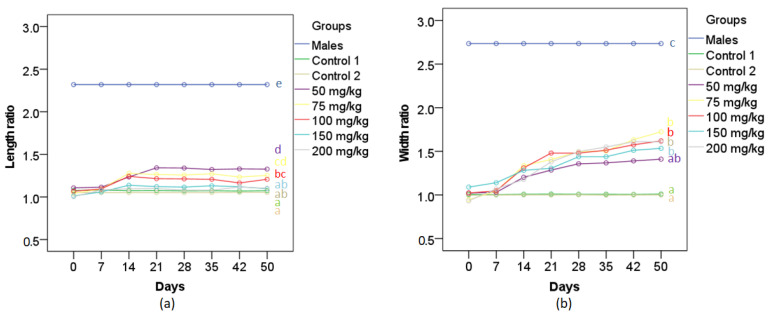
Temporal changes in (**a**) the ratios of the lengths of anal fin rays 4 and 6 and (**b**) the ratios of the widths of anal fins 3 and 4 of RG females treated with different concentrations of MT in their diet (*n* = 5). Lines with different letters (color coded to match those of the lines) are significantly different from one another.

**Figure 4 biology-11-00694-f004:**
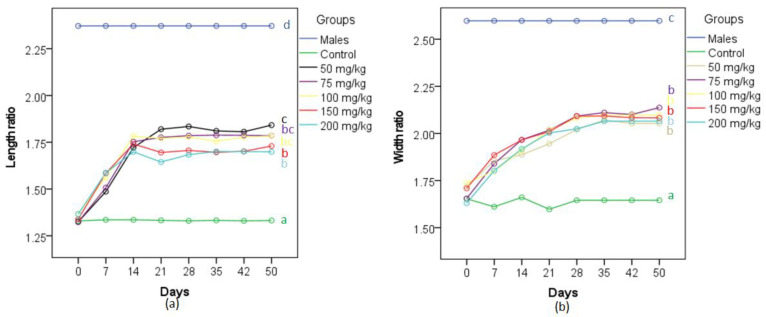
Temporal changes in (**a**) the ratios of the lengths of anal fin rays 4 and 6 and (**b**) the ratios of the widths of anal fin 3 and 4 of MG females treated with different concentrations of MT in their diet (*n* = 11). Lines with different letters (color coded to match those of the lines) are significantly different from one another.

**Figure 5 biology-11-00694-f005:**
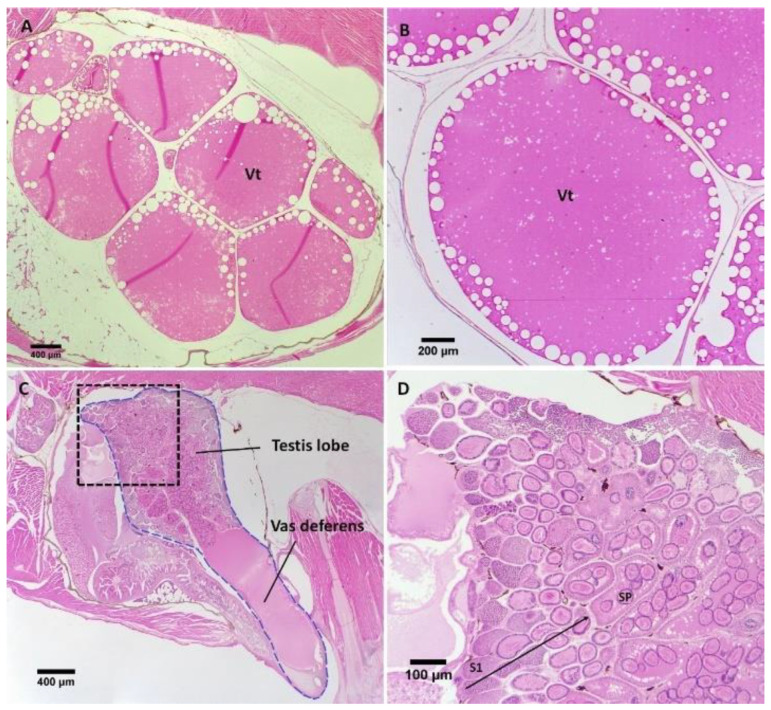
Sagittal sections through gonads of control adult female (**A**,**B**) and male (**C**,**D**) *G. holbrooki*. (**A**) Vitellogenic oocytes (Vt) typically occupied most part of ovarian cavity and (**B**) vitellogenic oocytes at higher magnification. (**C**) Testis (bound by the blue broken line) showing anterior glandular tissue and posterior vas deferens with (**D**) higher magnification (of the section framed by black broken line in (**C**) showing advancing stages of spermatogenesis from periphery towards central lumens (indicated by the direction of large arrow). Primary spermatocytes (S1) were located at the periphery of the testis while spermatozeugmata (SP) were presented in central lumens. Anterior to left and posterior to right.

**Figure 6 biology-11-00694-f006:**
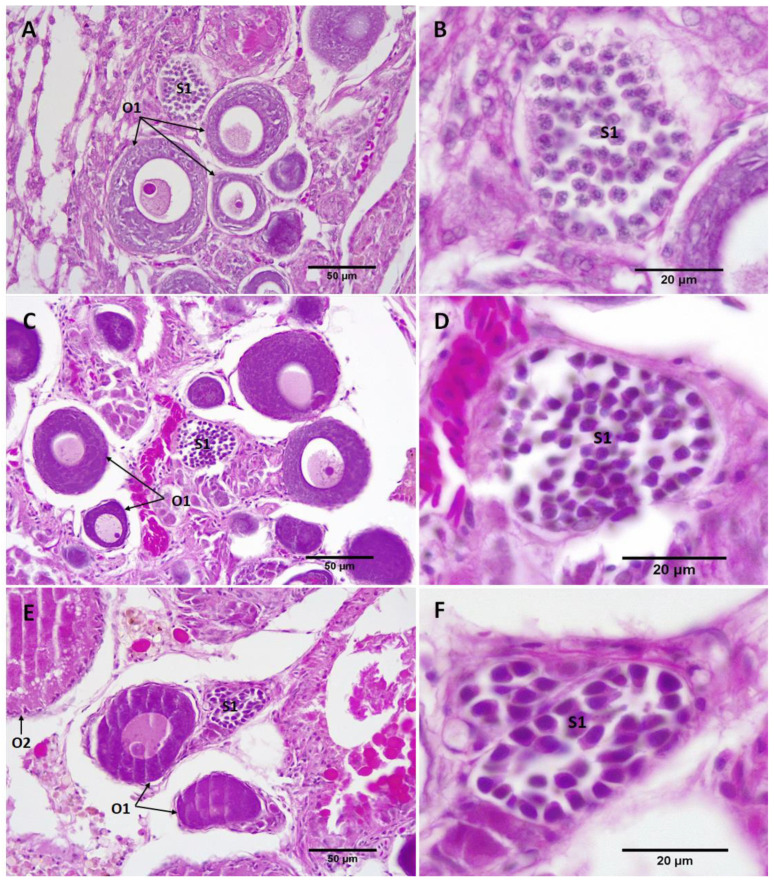
Sagittal sections through the gonads of MT-treated RG females showed the presence of primary spermatocytes in the ovary of MT-treated females. (**A**) The gonad of the female fed with 50 mg MT/kg diet and (**B**) its spermatocyte cyst at higher magnification; (**C**) gonad of the female fed with 75 mg MT/kg diet and (**D**) its spermatocyte cyst at higher magnification; and (**E**) gonad of the female fed with 150 mg MT/kg diet and (**F**) its spermatocyte cyst at higher magnification. O1: primary oocytes, O2: cortical alveoli oocytes, S1: primary spermatocytes.

**Figure 7 biology-11-00694-f007:**
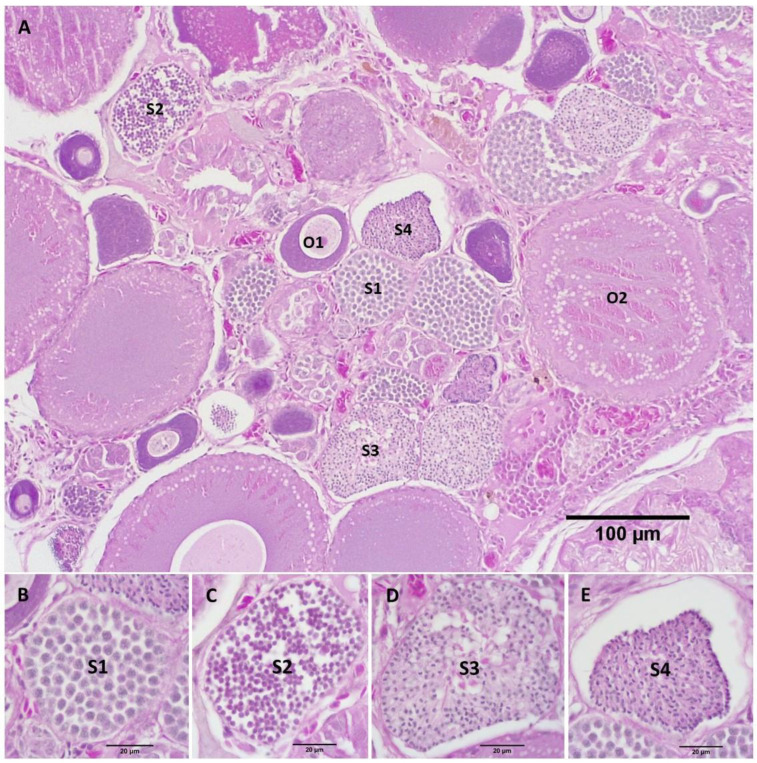
Sagittal sections through the gonad of a RG female fed with the 200 mg MT/kg diet. (**A**) The gonad presented several stages of follicular development from primary oocytes (O1) to cortical alveoli oocytes (O2) and various stages of spermatogenesis from primary spermatocytes (S1), secondary spermatocytes (S2) to early (S3) and late (S4) stages of spermatid. (**B**–**E**) The structures of S1, S2, S3, S4 at higher magnification, respectively.

**Figure 8 biology-11-00694-f008:**
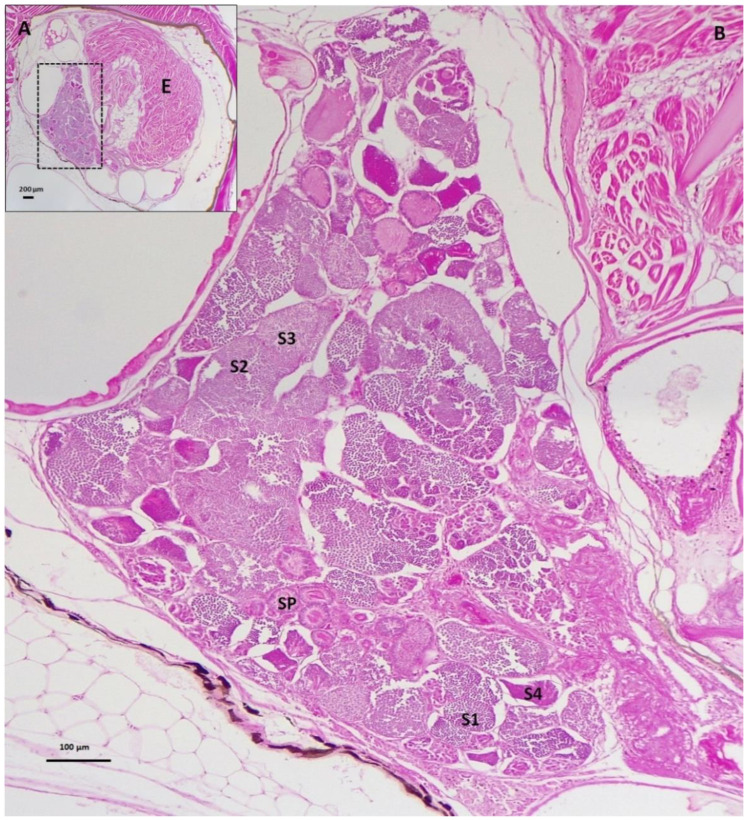
Sagittal sections through the gonad of the MG female treated with the 50 mg MT/kg diet. (A) Testicular tissue (inset framed by black broken rectangle) located next to an embryo (E) observed inside the ovarian sac of the treated female. (B) The testicular tissue at higher magnification showed various stages of spermatogenesis such as primary spermatocytes (S1), secondary spermatocytes (S2), early (S3) and late (S4) spermatids as well as spermatozoa in spermatozeugmata (SP). Anterior to left and posterior to right.

**Figure 9 biology-11-00694-f009:**
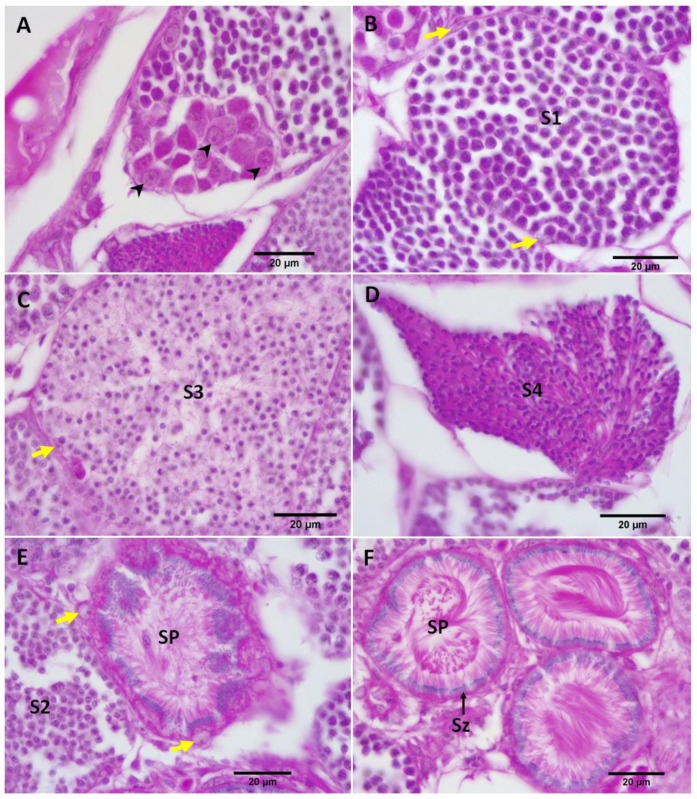
Sagittal sections through the gonad of the MG female treated with the 50 mg MT/kg diet at higher magnification: (**A**) spermatogonia (black arrowheads), (**B**) primary spermatocytes (S1), (**C**) early stage of spermatids (S3), (**D**) late stage of spermatids (S4), (**E**) secondary spermatocytes (S2) and early spermatozeugmata (SP) and (**F**) spermatozeugmata (SP) with the nuclei of the spermatozoa (Sz) oriented towards the periphery of spermatozeugmata. Sertoli cells are marked (yellow arrows).

**Figure 10 biology-11-00694-f010:**
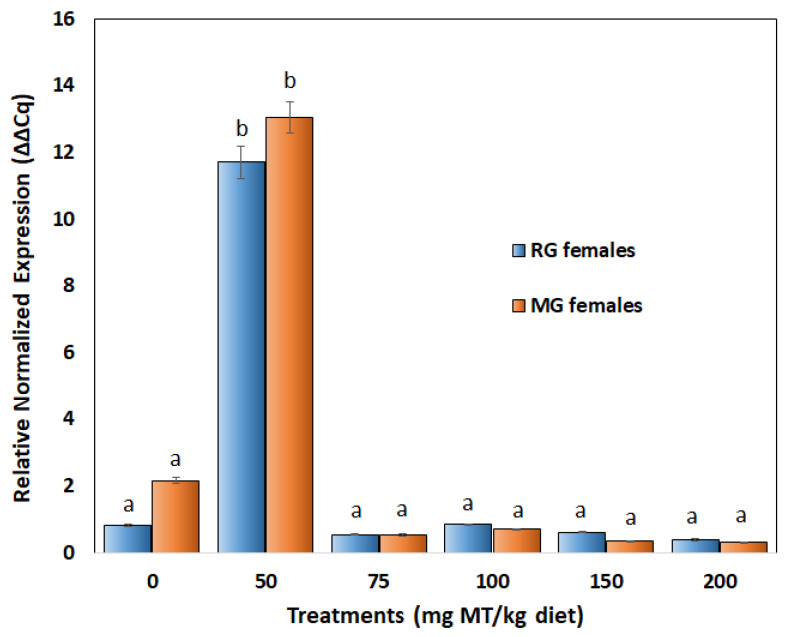
Relative normalized *amh* expression (∆∆Cq) in gonads of RG and MG females of *G. holbrooki* treated with MT (0 to 200 mg/kg). The data are presented as mean ± standard error (*n* = 3 per treatment). Means with different letters are significantly different from one another.

**Table 1 biology-11-00694-t001:** Total gonad and testicular tissue areas of RG females treated with MT for a 50-day period (*n* = 1 fish/treatment).

Treatment Doses	Tissue Area (µm^2^)		
(mg MT/kg Diet)	Total Gonad	Ovarian Tissue (%)	Testicular Tissue (%)
Control 1	21,544	21,544 (100)	0 (0)
Control 2	36,244	36,244 (100)	0 (0)
50	26,037	26,006 (99.9)	31 (0.1)
75	6902	6865 (99.5)	37 (0.5)
100	11,613	11,613 (100)	0 (0)
150	14,009	13,985 (99.8)	24 (0.2)
200	14,226	13,062 (91.8)	1164 (8.2)

**Table 2 biology-11-00694-t002:** Total gonad and testicular tissue areas of MG females treated with MT for a 50-day period (*n* = 1 fish/treatment).

Treatment Doses	Tissue Area (µm^2^)		
(mg MT/kg Diet)	Total Gonad	Ovarian Tissue (%)	Testicular Tissue (%)
Control female	22,790	22,790 (100)	0 (0)
Control male	30,113	0 (0)	30,113 (100)
50	16,274	8905 (55)	7369 (45)
75	11,347	11,347 (100)	0 (0)
100	16,450	16,450 (100)	0 (0)
150	5454	5454 (100)	0 (0)
200	18,599	18,599 (100)	0 (0)

**Table 3 biology-11-00694-t003:** The number of females with or without a gravid spot before and after MT treatment and those parturiated post treatment.

	Number with No Gravid Spot (%)	Number with Gravid Spot (%)	Number Parturiated (%)
Before treatment	23 (19.8)	93 (80.2)	- -
After treatment	116 (100)	0 (0)	2 (1.7)

## Data Availability

Not applicable.

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
