# Peer review of "Masculinization of Adult Gambusia holbrooki: A Case of Recapitulation of Protogyny in a Gonochorist?"

_biology, 2022, doi:10.3390/biology11050694_

Round 1

Reviewer 1 Report

The manuscript on masculinization of adult Gambusia holbrooki with 17α-Me- 2 thyltestosterone is well written, but it is too simple study with a prestigious title. I strongly recommend changing the title of the paper, it misleads the readers. Therefore, I recommend refining the content and scientific questions of the study to increase the novelty of the manuscript.

The main questions are: What is the benefit of sex reversal with MT in Gambusia for aquaculture?

In terms of conservation, the use of MT in natural environments is not useful. Is it possible for the authors to explain this?

What is the most important result of the study? There is no talk about the fact that this study provides the strong evidence of the plasticity of sex in this species. What is the useful result of the authors that could provide new knowledge about this species? For sex plasticity, it would have been much better to study known sex markers at different stages of gonadal development using quantitative real-time PCR and localization of known markers to understand this phenomenon at the molecular level. I recommend including some molecular studies as mentioned above and resubmitting the work. This species is an appropriate model organism for these types of studies. Accordingly, all sections of the study need to be justified by adding marker studies in the text.

The authors mentioned that this study can be the basis for the production of monosex offspring to control their pest populations. Is not it better to look at molecular aspects to address this question? How is it possible to produce monosexual offspring with MT on a large scale as in the natural environment? And then how to control their population in the environment? Further explanations are needed here.

Author Response

Dear reviewer,

Thank you for your valuable suggestions and comments. They have been very useful in revising the MS

Reviewer 2 Report

Review of: “Masculinization of adult Gambusia holbrooki using 17α-Methyltestosterone: A case of recapitulation of protogyny in a gonochorist” for biology (ID: 1623422).

Overview: This manuscript studied secondary sexual characteristics, internal gonad morphology, expression of amh gene and sexual behavior of the MT treatment mosquitofish at the two adult stages. I think there are a lot of holes in the experimental design, and the authors did not give good explanations of the significance and results of the experiment.

General:

  • What is the significance of this study? It is well established that hormones can induce sex change, and it is easy to understand why it is easier to do so early in life. Its efficacy on adults is meaningless to be studied. Even if, as described in the authors' experiment, adult females complete the sexual reversal, what is the productive significance of their existence? Can they reproduce?
  • The reason why the effect of MT at the dose of 50 mg/kg feed stimulated secondary sexual charter development better than that of 200 mg/kg diet is not clear, especially the only upregulated expression of amh in the gonads of females fed with 50 mg MT/kg diet. It is generally believed that the higher the hormone dose, the more pronounced the sex switch effect. What is the reason why low doses of hormones make sense?
  • I’m afraid that the wild specimens collected in the Tamar Island Wetland Reserve are not reliable. How did the authors determine that the fish collected were eastern mosquitofish (Gambusia holbrooki) rather than western mosquitofish (Gambusia affinis)? In addition, how were the authors sure the MG females? The sperm of male mosquitofish can be stored in the oviduct of female fish for a long time. It is difficult to ensure that fish caught in the wild have not been impregnated or mated. We don't think we can get an accurate estimate from body length.
  • I’m confused about the experimental design. For example, how many fish did the authors used in the experiment? What is the significance of setting the MT concentration gradient? Why the behavioral interactions only conducted in the females treated with 50 mg MT/kg diet?

Specific:

  • Line 14-22: Simple summary is not necessary.
  • Introduction is too long to get to the point.
  • Line 123-125: we’re quite confused about the logic of your aim in the preliminary experiments and your experimental design in this study.
  • Line 168-175: What did the condition factor (K) represent for? What does the high or low value reflect?
  • Line 180: why did the authors choose to measure the lengths of anal fin rays 4&6 and widths of anal fin rays 3&4?
  • Line 190: Only one female per treatment was sampled for histological examination is not reliable.
  • Line 127: wrong serial number
  • Figure 10: There was no different letters in the figure mentioned in the annotation.

Author Response

Dera reviewer,

Thank you for your suggestions and comment. Please see attached with detailed responses

Reviewer 3 Report

This study aimed to investigate how adult sex reversal occurs in hermaphroditic fish as a result of 17α-Methyltestosterone (MT) incorporated feed. Using eastern mosquitofish Gambusia holbrooki as a model organism, they found evidence of male copulatory structures and testicular tissue after oral administration of MT to mature females. They also tested the altered expression of one candidate sex differentiation gene, Anti-Müllerian Hormone (amh). They strengthened their inference by identifying altered sexual behavior among MT-treated females. Overall, this study provides evidence for sex-reverse in both maiden gravid and repeat gravid (RG) eastern mosquitofish females. 

Most of the fish biology and physiology results are convincing in this study, except for the limited sample sizes for gonad histology analysis (one female per treatment). However, I am concerned about the genetic part in this manuscript, given that this special issue is focused on genetics and genomics biotechnology.

Overall, the approach used to identify responses of genes involved in sex differentiation seems reasonable, and these are interesting gene expression patterns, but I simply don’t believe the interpretations offered. The gene expression changes of amh, analyzed in other paper, are so prevalent in fishes that perhaps it is not surprising for some to be coincident with identified pattern. Based on the existed literature, sex determination is a very complex trait, in which different key genes or sex-linked chromosomes were frequently recruited.

For G. holbrooki, a recent study that focused on the origin of sex determination mechanisms has identified “a unique allele of GIPC PDZ domain containing family member 1 (gipc1) on the Y chromosome (gipc1Y), which is linked to the potentially sexually antagonistic melanic phenotype”. (Kottler et al. 2021). In my mind, gene expression analyses should focus on a variety of sex-lined genes rather than a single gene that may interact with genetic of sex. The overall impression left with me by this paper is an exuberance for unusual gene expression change behind a single amh gene without skepticism and alternative hypotheses for what could be causing the patterns. For example, why only 50 mg MT/kg diet females have shown significant up regulation, but not in other MT concentration groups? Is there any difference between MT-treated RG versus MG females? 

Other, minor comments:

L102-113: given that the exact mechanism of sex-determination in G. holbrooki is still unclear, it would be helpful to introduce other potential genes linked to the sexually antagonistic phenotypes in this species. See Kottler et al. 2020, but also Geffroy et al 2021 for polygenic sex determination system hypothesis. 

L144: delete “i.e., small sized females (25 - 33 mm) 144 with no prior parturition history” - repeated contents.

L166: I am wondering if the temperature is involved in sex-reverse. Perhaps the authors could provide some description about how abiotic factors impact sex in this model species (at introduction part).

L190: I understand that it is a lot of work for gonad histology analysis, but only one female per treatment is still too small. Same for Table 1, it would be helpful to see variance across ovarian testicular tissue, even from different slides.

L526 Figure 10. Authors need to clarify what group this figure is presenting, MG or RG females?

Referece:

Geffroy, Benjamin, et al. "Unraveling the genotype by environment interaction in a thermosensitive fish with a polygenic sex determination system." Proceedings of the National Academy of Sciences 118.50 (2021).

Kottler, Verena A., et al. "Independent origin of XY and ZW sex determination mechanisms in mosquitofish sister species." Genetics 214.1 (2020): 193-209.

Author Response

Dear Reviewer,

Thank you for your valuable comments and suggestions. Please find attached detailed responses.

Round 2

Reviewer 1 Report

Dear Authors,

The revised version show some improvements but in my opinion remain clearly not publishable at this point. Several problems raised during the first round of review are still present and new ones appeared. It shows the lack of effort and/or interest of the authors to raised the manuscript at publishable level. Thus, I recommend the greater improvements on this manuscript as previously advised before considering for publication.

Author Response

Please see attached responses

Reviewer 2 Report

We have reviewed the revised manuscript. Although we still cannot agree with some views, the author has made great efforts in replying and revising them. We think the article can be published.

Reviewer 3 Report

The revised manuscript looks much better. It sounds causal to pool MG and RG-treated females for QPCR analyses without skepticism and alternative hypotheses. I would recommend authors split them for data visualization and discuss potential mechanisms for why there is no significant difference between MG and RG females.  
